# UPLC–MS/MS and Gene Expression Research to Distinguish the Colour Differences of *Rhododendron liliiflorum* H. Lév

Jin Dai [1,†], Xinglin Wang [1,†], Xingpan Meng [1], Xu Zhang [2], Qihang Zhou [3], Zhengdong Zhang [4], Ximin Zhang [1], Yin Yi [1], Lunxian Liu [1,*] and Tie Shen [1,3,*]

1 Key Laboratory of National Forestry and Grassland Administration on Biodiversity Conservation in Karst Mountainous Areas of Southwestern China, Engineering Research Center of Carbon Neutrality in Karst Areas, Ministry of Education, Key Laboratory of Environment Friendly Management on High Altitude Rhododendron Diseases and Pests, Institutions of Higher Learning in Guizhou Province, School of Life Science, Guizhou Normal University, Guiyang 550025, China; 21010100373@gznu.edu.cn (J.D.); 232100100407@gznu.edu.cn (X.W.); 222100100438@gznu.edu.cn (X.M.); zhxm409@gznu.edu.cn (X.Z.); gzklppdr@gznu.edu.cn (Y.Y.)
2 Guizhou Caohai Wetland Ecosystem National Observation and Research Station, Guizhou Academy of Forestry Sciences, Guiyang 550001, China; lkyzx163@163.com
3 Key Laboratory of Information and Computing Science Guizhou Province, School of Mathematical Sciences, Guizhou Normal University, Guizhou 550025, China; 19010210498@gznu.edu.cn
4 College of Mathematics and Information Science, Guiyang University, Guiyang 550001, China; zzd@gznu.edu.cn
* Correspondence: llx@gznu.edu.cn (L.L.); shentie@gznu.edu.cn (T.S.)
† These authors contributed equally to this work.

**Abstract:** Among ornamental plants, the colour of the petals is an important feature. However, the reason for the colour differences of *Rhododendron liliiflorum* remains unclear. To reveal the differences in the colour of *R. liliiflorum*, high-efficiency liquid chromatographic collar (UPLC–MS/MS) technology was used to study the yellow and white parts of *R. liliiflorum*. A total of 1187 metabolites were identified in *R. liliiflorum* petals, including 339 flavonoid metabolites. Seventy-eight types of flavonoids in these metabolites were found in the yellow and white parts of *R. liliiflorum* petals, along with 11 other significantly enriched substances. Combining gene expression-related data with differential metabolite data demonstrated effects of enrichment in the flavanonols (fustin), flavonols (epiafzelechin and afzelechin), and flavanones (pinocembrin) of flavonoid biosynthesis; glyccitin, 6″-O-malonylgenistin, and 6-hydroxydaidzein of isoflavonoid biosynthesis; and anthocyanin biosynthesis of malvidin-3-O-galactoside (primulin), delphinidin-3-O-rutinoside, cyanidin-3-O-glucoside (kuromanin), and cyanidin-3-O-rutinoside (keracyanin), which are potentially the contributing factors responsible for the differences in petal colour in *R. liliiflorum*. This study establishes a connection between the differential metabolites underlying the color differences in the petals of *R. liliiflorum* and the gene expression in *R. liliiflorum*. This will provide a foundation for subsequent research on the regulation of flower color in *R. liliiflorum* and have profound implications for horticultural applications of *R. liliiflorum*.

**Keywords:** colouration; petal; metabolites; flavonoids

## 1. Introduction

*Rhododendron* is the largest genus of the Rhododendron family, with wild rhododendrons found all over the world [1]. There are traces of azalea discovered in low latitudes from tropical areas to high latitudes. Wild *Rhododendron* genus communities have contributed to the economic development of local communities, such as the Baili Rhododendron Nature Reserve in northwestern Guizhou Province, China [2]. Rhododendron flower colour is highly diverse, from purple to red, pink, and even white [3]; thus, its extensive usage in research on plant flower colours [4–6]. *Rhododendron Liliiflorum* belongs to the Rhododendron family and

exists in southeast China (southeast of Yunnan to Hunan). The base of the petals in the late flowering period is yellow and the upper part is white (Figure 1a,b).

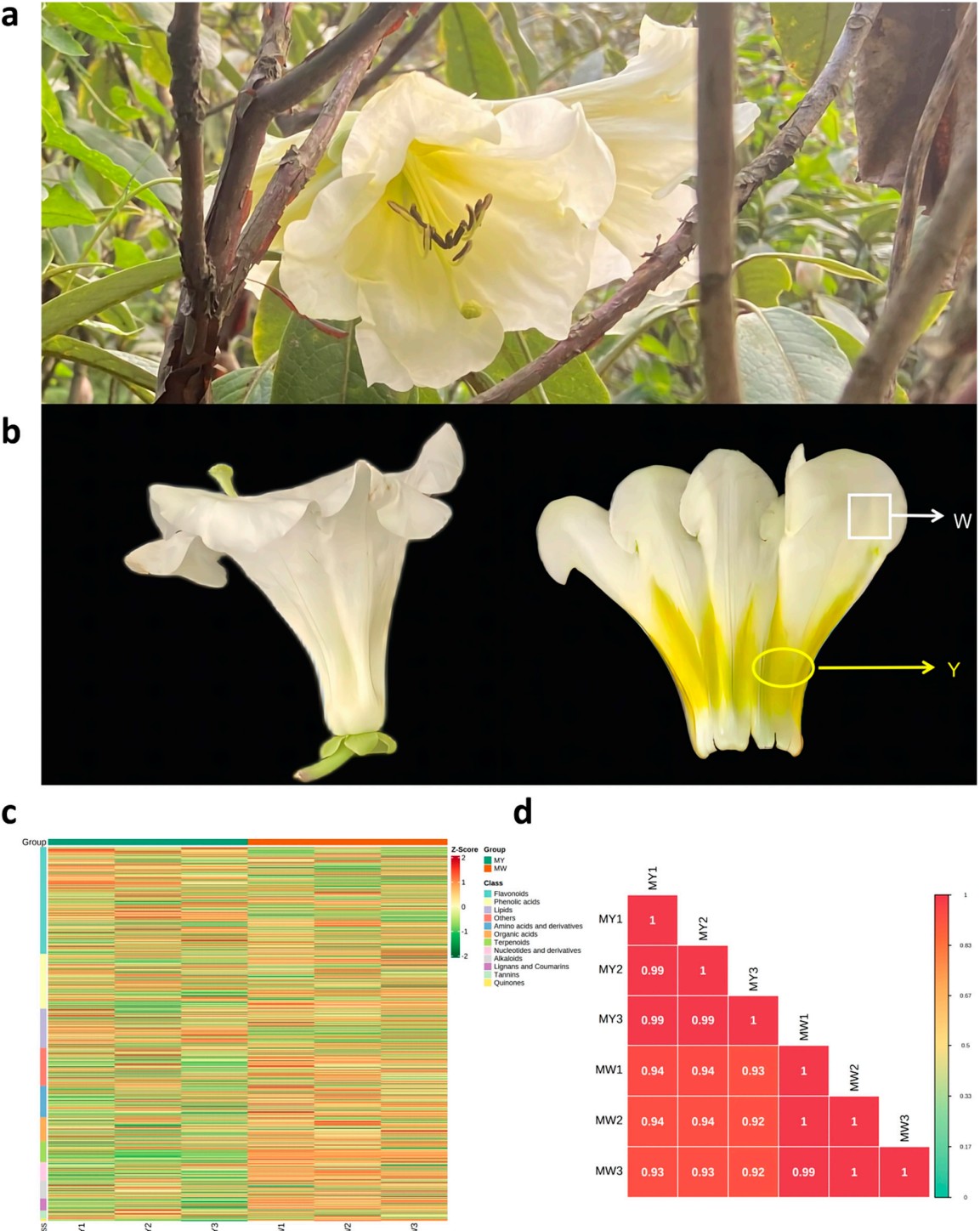

**Figure 1.** *R. liliiflorum* phenotype and overall characteristics. (**a**). Wild *R. liliiflorum.* (**b**). The morphology of *R. liliiflorum*, *R. liliiflorum* petals overall diagram, *R. liliiflorum* petal inner diagram. The part marked by the white square border is the white part of the *R. liliiflorum* (W). The yellow elliptical border bezel. The marked part is the yellow part (Y) of *R. liliiflorum*. (**c**). MWs vs. MYs overall clustering heatmap; horizontal is sample name, longitudinal information is metabolites, groups are packets, and class classification is material. (**d**). MW vs. MY correlation analysis diagram.

Among ornamental plants, the colour of the petals is an important feature. Studies have shown that colour characteristics are an important factor in the interaction between plants and their powder [7,8]. The potential factors affecting the colour of plants include hydrogen, metal ions, pigment ingredients, and petal structure [9]. The most critical factor is pigment composition. The pigment ingredients that affect the formation of colours include flavonoids, carotene, betaine, etc. Each of these combinations adds to the diversity of colour [10]. Flavonoids are a class of polyphenol secondary metabolites, which are natural pigments widely present in plants. Based on the phenolic molecular structure with the heterocyclic ring and conformation, these metabolites can be divided into flavones, flavonols, isoflavonoids, flavanones, flavanols, anthocyanidins, and so on [11–13]. Flavonoids are widely present in plants, and their contents in flowers [14], leaves [15], fruits [16], and other parts of plants are high. The colour changes of petals may be caused by different flavonoids, while the proportional changes of various types of flavonoids may be a pertinent factor in the change in colour. This includes anthocyanidins, flavanones, chalcones, flavonols, isoflavonoids, and flavanones [17,18]. In a study of lotus petals, it was found that the content of anthocyanins was positively correlated with red petals, and flavonols contributed to the yellow part of the auxiliary pigment [19]. After comparing seven types of plants with red flowers and white flowers, it was found that the content of anthocyanins in safflower was significantly higher than that in white flowers. In white flowers, flavonol is the main pigment type [20].

As discovered through many studies, anthocyanins, a downstream material for flavonoid biosynthesis, is considered to be the main factor stimulating colour changes in plant flowers [21]. For example, a study by Mizuta et al. [22] on the pattern of flower colour and the anthocyanin composition of evergreen rhododendrons found that purple rhododendrons had more anthocyanins in the red, purple, and white groups of samples, while no anthocyanin content was detected in white rhododendrons [22]. Sun et al. [23] used 'Yanzhi Mi' (pink azalea) and the wild-type (WT) cultivar 'Dayuanyangjin' (white azalea with pink stripes) as their study subjects and found that pink-petalled azalea showed higher anthocyanin contents, while white-petalled azalea contained mainly flavonoids and a small amount of anthocyanin, which may be attributable to the presence of stripes on white-petalled azalea [24]. In a study of the wild type of primrose (yellow petals) and its variants (yellow transformed to red), it was found that the orange and red primrose types contain anthocyanins such as cyanidin-3-O-glucoside [23]. Most of the current research on rhododendrons focuses on white rhododendrons and red rhododendrons, and many studies on white rhododendrons have shown that white rhododendrons have lower anthocyanin contents [22,24,25]. Additionally, the enzyme genes CHS, F3'5'H, FLS, I2'H, HID, DFR, and LAR may have a role in the alteration of petal colour in *R. liliiflorum* [21,26–28]. However, few studies have revealed colour differences within the same petal colour.

Metabolomics is widely used in research on plant colour formation [25,29,30]. Metabolic group studies have studied four noncolours of wheat, explaining that the reason for the colour difference of wheat is potentially the accumulation of flavonoid compounds [31]. Guan et al. [30] investigated the differentiation mechanism of the colouration of sophora flowers and identified the key metabolite responsible for the red colour as anthocyanins of the delphinidin type using metabolomics approaches. Combining the related genes of plant traits can reflect the reasons for changes in the organism. At present, related research on the colour metabolic pathway has been combined with metabolic gene research. For example, Wang et al. [31] combined metabolic group and gene expression analyses and explored phenylpropanoid biosynthesis and flavonoid biosynthesis in wheat of different colours. Jiang [32] comprehensively analysed anthocyanins in salvin.

Previously, we used the transcriptome to study *R. liliiflorum*, and the differentially expressed genes of the two parts of *R. liliiflorum* petals were selected. The expression of related enzymes may affect the difference in the colour of *R. liliiflorum* [21]. This study used high-efficiency liquid chromatography–coupled mass spectrometry (UPLC–MS/MS) to study the two parts of *R. liliiflorum,* yellow and white, and screened differentiated

metabolites during biological synthesis, such as phenylpropanoids, flavonoids, and anthocyanins. Gene expression was verified to better explain the formation mechanism of *R. liliiflorum* colour.

## 2. Materials and Methods

### 2.1. Plant Materials

Wild *R. liliiflorum* (Figure 1a) was picked in the Baili Rhododendron Nature Reserve (above sea level: 1060~2200 m; precipitation: 1.6 °C; mean temperature of air: 1180.8 mm; N 27°12′54″, E 105°55′5″) in northwestern Guizhou Province and was subsequently planted in the nursery of Guizhou Normal University (above sea level: 1100 m; precipitation: 24 °C; mean temperature of air: 1129.5 mm; N 26°35′18″, E 106°43′18″). In April 2021, fresh *R. liliiflorum* was collected from the nursery of Guizhou Normal University. After collection, each *R. liliiflorum* petal was divided into two parts: yellow (Y) and white (W). Subsequently, the samples were quickly placed in liquid nitrogen and frozen at −80 °C to provide experimental materials for subsequent experiments.

### 2.2. Experimental Method

#### 2.2.1. Sample Preparation and Extraction

This experiment divided *R. liliiflorum* into white (W) and yellow (Y) parts, divided into two groups (MW, MY), and every group of samples had 3 biological repeats. Biological samples were freeze-dried with a vacuum freeze-dryer (Scientz-100F). Freeze-dried samples were crushed using a mixer mill (MM 400, Retsch, Shanghai, China) with a zirconia bead for 1.5 min at 30 Hz. Then, 100 mg of lyophilized powder was dissolved in 1.2 mL of a 70% methanol solution, vortexed for 30 s every 30 min 6 times in total, and placed in a refrigerator at 4 °C overnight. Following centrifugation at $12,000\times g$ rpm for 10 min, the extracts were filtered (SCAA-104, 0.22 µm pore size; ANPEL, Shanghai, China) before UPLC–MS/MS analysis.

#### 2.2.2. UPLC Conditions

The sample extracts were analysed using a UPLC–ESI–MS/MS system (UPLC, SHIMADZU Nexera X2; MS, Applied Biosystems 4500 Q TRAP). The analytical conditions were as follows: UPLC—Agilent SB-C18 column (1.8 µm, 2.1 mm × 100 mm). The mobile phase consisted of solvent A, pure water with 0.1% formic acid, and solvent B, acetonitrile with 0.1% formic acid. Sample measurements were performed with a gradient programme that employed the starting conditions of 95% A, 5% B. Within 9 min, a linear gradient to 5% A, 95% B was programmed, and a composition of 5% A, 95% B was kept for 1 min. Subsequently, a composition of 95% A and 5.0% B was adjusted within 1.1 min and kept for 2.9 min. The flow velocity was set as 0.35 mL per minute, the column oven was set to 40 °C, and the injection volume was 4 µL.

#### 2.2.3. ESI-Q TRAP-MS/MS

LIT and triple quadrupole (QQQ) scans were acquired on a triple quadrupole-linear ion trap mass spectrometer (Q TRAP) of the AB4500 Q TRAP UPLC/MS/MS system, equipped with an ESI turbo ion spray interface operating in positive and negative ion mode, and controlled by Analyst 1.6.3 software (AB Sciex). The ESI source operation parameters were as follows: ion source, turbo spray; source temperature, 550 °C; ion spray voltage (IS), 5500 V (positive ion mode)/−4500 V (negative ion mode); ion source gas I (GSI), gas II (GSII), and curtain gas (CUR) set at 50, 60, and 25.0 psi, respectively; and collision-activated dissociation (CAD), high. Instrument tuning and mass calibration were performed with 10 and 100 µmol/L polypropylene glycol solutions in QQQ and LIT modes, respectively. QQQ scans were acquired as MRM experiments with collision gas (nitrogen) set to medium. DP and CE for individual MRM transitions were performed with further DP and CE optimization. A specific set of MRM transitions was monitored for each period according to the metabolites eluted within this period.

2.2.4. Data Quality Control and Statistics Analysis

To test the reproducibility of the samples during the extraction and detection process, we performed and obtained quality control (QC) on the samples; we then analysed the occurrence frequency of the CV (coefficient of variation) for substances smaller than the reference value using the empirical cumulative distribution function (ECDF). After showing the stability of the experimental data, we first used R software's (www.r-project.org/; accessed on 15 July 2021) built-in statistics for the two sets of samples to perform the main component analysis method (PCA). The structure compared the differences between the yellow and white parts of the *R. liliiflorum* petal and the three parallel experimental groups. Using the software R Metaboanalystr (V1.0.1) to perform orthogonal signal correction and partial two-multiplication analysis, the OPLS-DA model was established by the *R. liliiflorum* yellow and white parts (orthogonal bias minimum multiplication judgement analysis). The metabolic content data used normalized processing (UV Scaling), and the clustering heatmaps were drawn using R software's Complexheatmap (V 2.8.0).

To further find the cause of the colour differences between the white part (W) and the yellow (Y) section of *R. liliiflorum*, taking |log2fold change| $\geq$ 1 and VIP $\geq$ 1 as the screening conditions, the different metabolites were screened using the KEGG Compound database (http://www.kegg.jp/kegg/compound/; accessed on 17 July 2021) and then the annotated metabolites were mapped to the KEGG Pathway database (http://www. kegg.jp/kegg/pathway.html; accessed on 19 July 2021). Whereafter, using a Pearson's correlation coefficient (|r|) greater than 0.8 and $p < 0.05$ as the screening conditions, the difference between the yellow and white parts of the R. lili-iflorum showed the correlation between the screened material. Correlation analysis can help determine the metabolic closeness between metabolic proximities, which is conducive to further understanding the mutually regulating relationship between metabolites. Common analysis of genes and metabolic groups represents the correlation between metabolites and genes through network diagrams. Regarding Pearson's correlation coefficient (|r|), the closer it was to 1, the stronger the correlation. We used |r| > 0.8. Unigene and different metabolites to draw in the flavonoid, anthocyanin, and isoflavonoid biosyn-thesis. Finally, the pathway of flavonoid biosynthesis of differential metabolites was obtained. According to the front and back relationships of flavonoids, we used enzymes that are closely related to flavonoids as intermediaries and used in plants to synthesize the path diagram.

**3. Results**

*3.1. Morphological Characteristics and Overall Characteristics of Metabolites*

The wild *R. liliiflorum* is shown in Figure 1a. The upper base part of the *R. liliiflorum* petals was yellow and white. This experiment divided *R. liliiflorum* into the white part (MW) and yellow part (MY) (Figure 1b). Using principal component analysis (PCA), the sample was divided into three parts: *R. liliiflorum* petals, yellow and white parts, and quality control samples. The quality control samples were prepared by mixing them with sample extracts. PCA scoring graphs are shown in Figure S1A. PC1 and PC2 explained 42.00% and 21.00% of the variance in the total samples, which effectively separated the yellow and white parts. The cover inspection of the quality control (QC) samples had a good overlap, which proved that the experimental conditions were stable. In this study, we built an OPLS-DA model for the white part (MW) and yellow part (MY) of *R. liliiflorum* (Figure S1B). OPLS-DA is a diverse statistical analysis method with supervision mode recognition. It can effectively eliminate the impact, which is irrelevant to research screening the differences in differential metabolites. In this model, $R^2X$ and $R^2Y$ represented the interpretation rate of the X and Y matrix of the models built and $Q^2$ represented the predictive ability of the model. The closer the matrixes to the indicators of $R^2X$ and $R^2Y$, the more reliable the model, and $Q^2$ was greater than 0.9, so the model was very good. From the verification diagram (Figure S1B), it is demonstrated that $Q^2 = 0.966$, $R^2X = 0.668$, and $R^2Y = 1$ for the yellow part (MY), and the white part (MW) indicated that the OPLS-DA model established by the *R. liliiflorum* was good. The prediction ability was reliable. After performing the overall process of the

metabolite data analysis, the difference in the accumulation of metabolites in the yellow and white parts of *R. liliiflorum* petals was displayed through heatmaps. The results showed (Figure 1c) that the material classification of the two parts of *R. liliiflorum,* yellow and white, was concentrated in more flavonoids, followed by phenolic acids, lipids, others, amino acids, and derivatives. Figure 1d shows that the three groups of samples of the white and yellow parts of the *R. liliiflorum* petals and the yellow parts |r| were greater than 0.9.

### 3.2. Identification and Comparison of Metabolites and Differentiated Metabolites

To better understand the changes in metabolites in the two parts of *R. liliiflorum* petals, we identified the primary metabolites and secondary metabolites in the sample through the database established by the UPLC–MS/MS. After identification, a total of 1187 types of metabolites were detected, including 339 kinds of flavonoids, 177 kinds of phenolic acids, 124 varieties of lipids, 100 amino acids and derivatives, 75 organic acids, 67 terpenoids, 58 nucleotides and derivatives, 57 kinds of alkaloids, 38 kinds of lignans and coumarins, 26 tannins, 7 kinds of quinones, and 119 types of other types (Table 1). We found that the detected flavonoid metabolites were the most abundant. Among the flavonoid metabolites were chalcone, flavanones, flavanonols, flavones, flavonols, flavonoid carbonoside, flavanols, anthocyanidins, isoflavones, and dihydroisoflavones. There were 20, 44, 13, 54, 122, 16, 23, 27, 18, and 2 species, respectively.

**Table 1.** Flavonoid metabolite secondary classification.

| Secondary Classification of Flavonoids | Quantity | Percentage (%) |
| --- | --- | --- |
| Chalcone | 20 | 5.8% |
| Flavanones | 44 | 12.9% |
| Flavanonols | 13 | 3.8% |
| Flavones | 54 | 15.9% |
| Flavonols | 122 | 35.9% |
| Flavonoid carbonoside | 16 | 4.7% |
| Flavanols | 23 | 6.7% |
| Anthocyanidins | 27 | 7.9% |
| Isoflavones | 18 | 5.3% |
| Dihydroisoflavones | 2 | 0.5% |

Note: A total of 339 flavonoid metabolites were screened.

### 3.3. Flavonoid Differential Metabolites

With |Log2Fold Change| $\geq$ 1, VIP $\geq$ 1 was used as a condition to screen the differential metabolites (Figure 2a). The results demonstrated that the comparison of the white part (MW) and yellow part (My) consisted of a total of 197 different metabolites, 78 types of flavonoids, and metabolites (Table S1). Of these, 82 differential metabolites were upregulated and 31 flavonoid-differentiated metabolite expressions were upregulated. There were 115 differential metabolites that were downregulated and 31 flavonoid-differentiated metabolite expressions were downregulated.

The 78 metabolites were flavanones, chalcones, flavanonols, anthocyanidins, flavones, isoflavones, flavonols, flavanols, and flavonoid carbonoside, of which there were 5, 4, 1, 12, 9, 7, 26, 8, and 6 of each type. Among the substances that were screened, some of the differences were detected in the yellow flower part of *R. liliiflorum* (Figure 2b), such as chalcone isosalipurposide-6″-O-p-coumaric acid, anthocyanidin delphinidin-3-O-rutinoside, flavone baicalein and chrysoeriol-7-O-(6″-malonyl)glucoside, isoflavone calycosin-7-O-glucoside, 6″-O-malonylgenistin, and 6-hydroxydaidzein. Flavonoid carbonoside chrysoeriol-6-C-glucoside-4′-O-glucoside, flavonol 6-C-methylquercetin-3-O-rutinoside, isorhamnetin-3-O-neohesperidoside, isorhamnetin-3-O-galactoide-7-O-rhamnoside, and isorhamnetin-3-O-glucoside-7-O-rhamnoside were also detected.

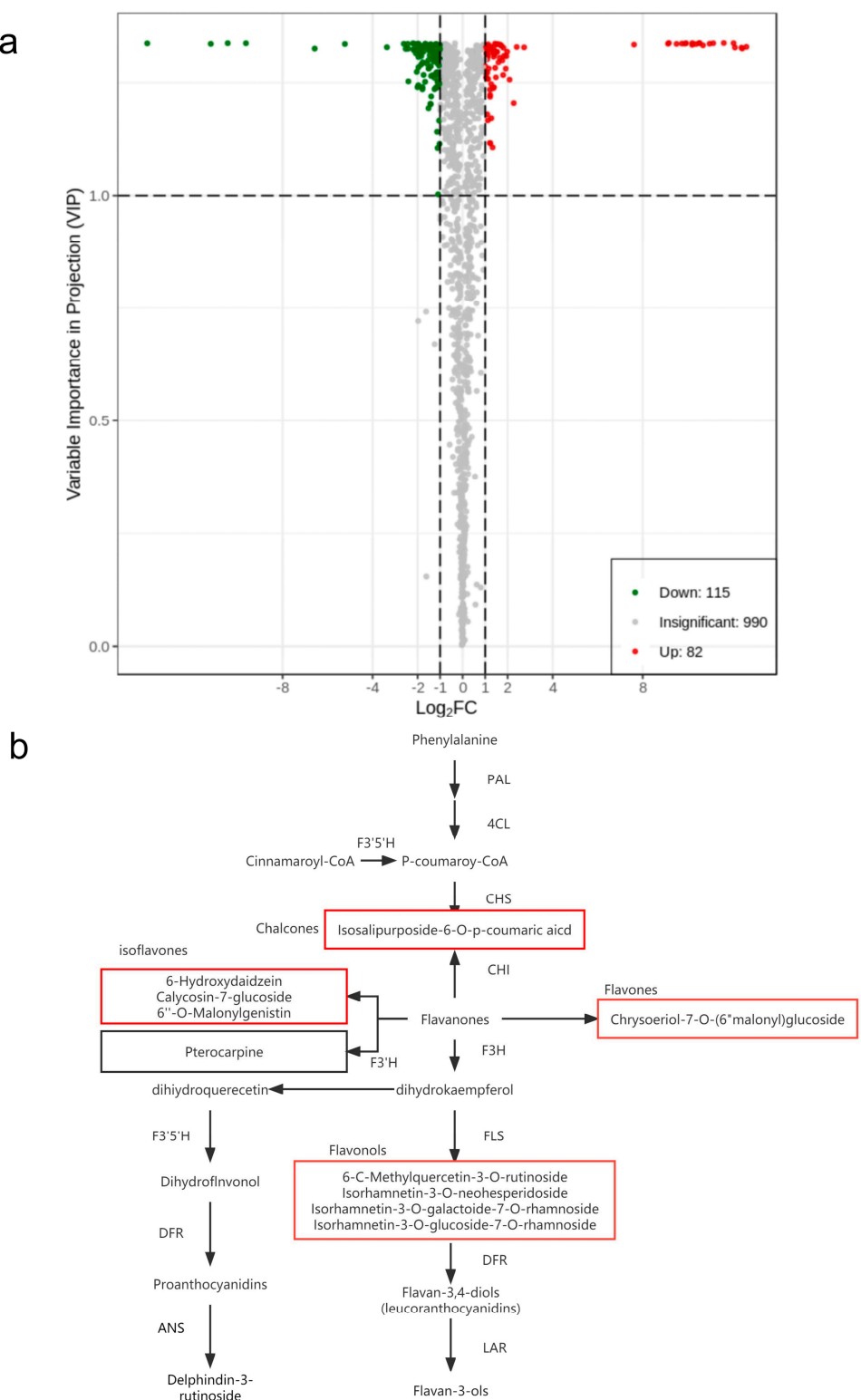

**Figure 2.** (**a**). Volcanic map of the differences in metabolites in *R. liliiflorum* flowers in the yellow (Y) and white (W) parts. Red dots represent the differential metabolites, green dots represent the differential metabolites, and grey represents the differences. (**b**). The flavonoid synthesis chart shows the specific differences in the white or yellow parts of *R. liliiflorum*. The substance in the black square was significantly expressed in the white part. The substance in the red square was significantly expressed in the yellow part.

*3.4. KEGG Enrichment Analysis of Metabolites*

First, we used │Log2Fold Change│ ≥ 1 and VIP ≥ 1 to screen metabolites and match the selected differential metabolites with the KEGG database to obtain the annotation path of the differential metabolites. Subsequently, we chose the top 20 prominent pathways and then made bubble drawings (Figure 3). We found that the differences between the yellow parts of *R. liliiflorum* and the white part were mainly enriched in isoflavonoid biosynthesis, flavonoid biosynthesis, and anthocyanin biosynthesis (Table 2). Here, we found that six different metabolites were enriched in flavonoid biosynthesis, namely, with pinocembrin (dihydrochrysin) in flavanones, epiafzelechin and afzelechin (3,5,7,4′-tetrahydroxyflavan) in flavanols, fustin in flavanonols, and 5-o-caffeoyl shikimic acid and chlorogenic acid (3-O-caffeoylquinic acid) in phenolic acids. Four different metabolites were enriched in anthocyanin biosynthesis, namely, cyanidin-3-O-glucoside (kuromanin), malvidin-3-O-glucoside (oenin), cyanidin-3-O-rutinoside (keracyanin), and delphinidin-3-O-rutinoside. 6′-O-malonylgenistin, glycitin (glycitein 7-O-glucoside), and 6-hydroxydaidzein were enriched in isoflavonoid biosynthesis.

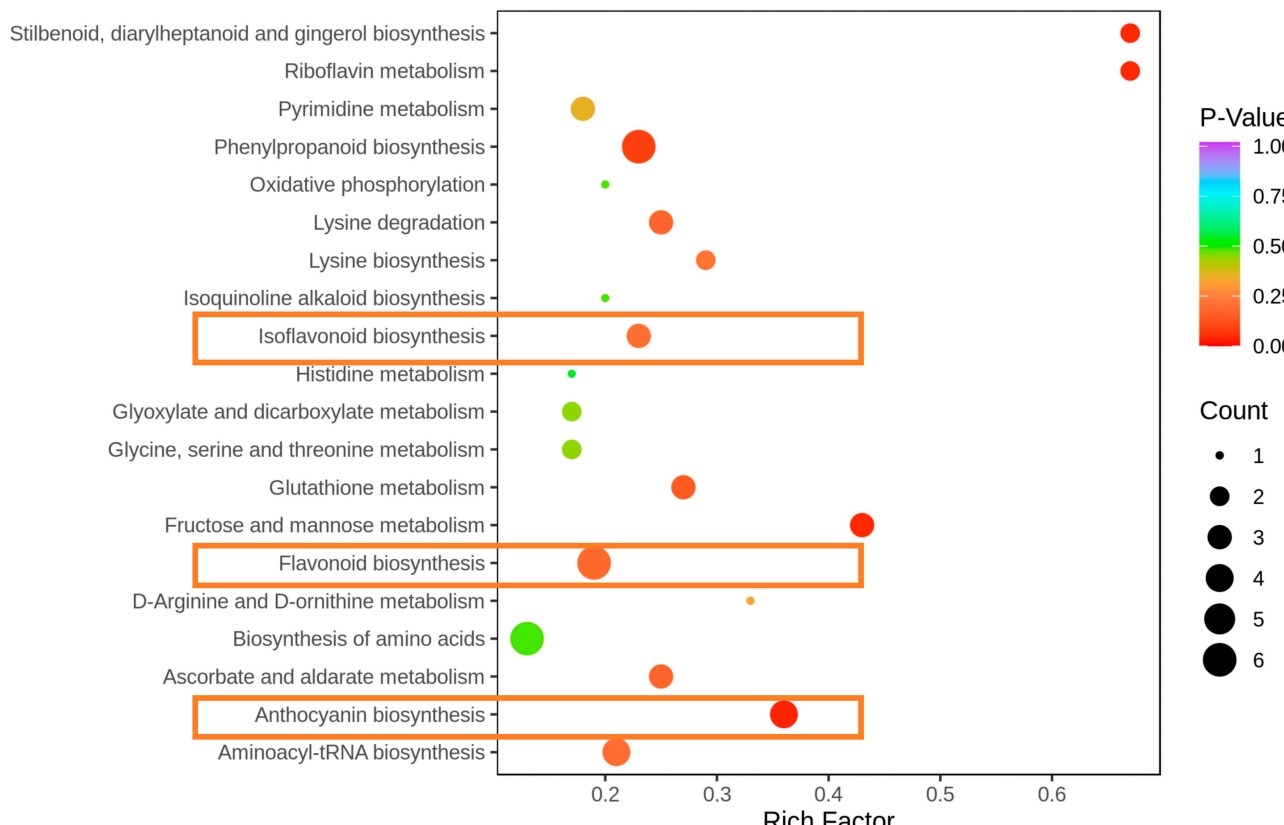

**Figure 3.** Kyoto Encyclopedia of Genes and Genomes (KEGG) pathway enrichment analysis of differentially expressed metabolites. The horizontal coordinate indicates the rich factor corresponding to each path. The vertical coordinate is the KEGG name. The greater the rich factor is, the greater the degree of enrichment. The larger the point, the greater the number of metabolites that are enriched in this way. The redder the colour of the dot, the more obvious the degree of enrichment. The yellow box represents the significantly enriched metabolic pathway that we are particularly interested in.

**Table 2.** The metabolites that are enriched in the KEGG pathway.

| Formula | Compounds | Class II | Type | CAS |
|---|---|---|---|---|
| $C_{15}H_{12}O_4$ | Pinocembrin (Dihydrochrysin) | Flavanones | down | 480-39-7 |
| $C_{15}H_{10}O_5$ | 6-Hydroxydaidzein | Isoflavones | up | 17817-31-1 |
| $C_{15}H_{14}O_5$ | Epiafzelechin | Flavanols | up | 24808-04-6 |
| $C_{15}H_{14}O_5$ | Afzelechin (3,5,7,4′-Tetrahydroxyflavan) | Flavanols | up | 2545-00-8 |
| $C_{15}H_{12}O_6$ | Fustin | Flavanonols | up | 20725-03-5 |
| $C_{22}H_{22}O_{10}$ | Glycitin (Glycitein 7-O-Glucoside) | Isoflavones | down | 40246-10-4 |
| $C_{21}H_{21}O_{11}+$ | Cyanidin-3-O-glucoside (Kuromanin) | Anthocyanidins | down | 47705-70-4 |
| $C_{23}H_{25}O_{12}+$ | Malvidin-3-O-glucoside (Oenin) | Anthocyanidins | down | 18470-06-9 |
| $C_{24}H_{22}O_{13}$ | 6″-O-Malonylgenistin | Isoflavones | up | 51011-05-3 |
| $C_{27}H_{31}O_{15}+$ | Cyanidin-3-O-rutinoside (Keracyanin) | Anthocyanidins | up | 28338-59-2 |
| $C_{27}H_{31}O_{16}+$ | Delphinidin-3-O-rutinoside | Anthocyanidins | up | 15674-58-5 |
| $C_{16}H_{16}O_8$ | 5-O-Caffeoylshikimic acid | Phenolic acids | down | 180981-12-8 |
| $C_{16}H_{18}O_9$ | Chlorogenic acid (3-O-Caffeoylquinic acid) | Phenolic acids | down | 327-97-9 |

Note: + represents the substance as a positive optical substance. Down represents the content of the substance in the white part (W) of *R. liliiflorum* compared to the yellow part (Y). UP indicates that the content of the species in the white part of *R. liliiflorum* (W) was lower than that in the yellow part (Y).

### 3.5. Flavonoid, Isoflavone, Anthocyanin Biosynthesis-Related, and Metabolite-Related Networks and Metabolite and Gene Correlation Networks

Eleven types of metabolites were filtered out, including alkaloids, amino acids and derivatives, flavonoids, lignans and coumarins, lipids, nucleotides and derivatives, organic acids, phenolic acids, quinones, tannins, and terpenoids (Figure 4). These 11 types of substances included 196 different metabolites, of which 78 types of flavonoids had different metabolites. The 78 types of flavonoids had a strong correlation. Four different metabolites were rich in flavonoid biosynthesis. Pinocembrin (dihydrochrysin) and epiafzelechin, afzelechin (3,5,7,4′-testrahydrovone), and fustin were negatively related. Pinocembrin (dihydrochrysin) decreased in the yellow part of the *R. liliiflorum* petals and epiafzelechin, afzelechin (3,5,7,4′-testrahydrovan), and fustin increased in the yellow part. The changes in correlations and contents were consistent. In anthocyanin biosynthesis, malvidin-3-O-glucoside (oenin), cyanidin-3-O-glucoside (kuromanin), delphinidin-3-O-rutinoside, and cyanidin-3-O-rutinoside (keracyanin) presented negative correlations, malvidin-3-O-glucoside (oenin) and cyanidin-3-O-glucoside (kuromanin) decreased in the yellow part of *R. liliiflorum* petals, and delphinidin-3-O-rutinoside and cyanidin-3-O-rutinoside (keracyanin) increased in the yellow part. The changes in correlations and contents were consistent. In the isoflavone synthesis pathway, the 6′-o-malonyl genistin and 6-hydroxydaidzein contents in the yellow part of *R. liliiflorum* were higher than those in the white part, and the Glycitin (glycitein 7-O-glucoside) content in the yellow part was lower than that in the white part. 6″-O-Malonylgenistin, 6-hydroxydaidzein, and glycitin (glycitein 7-O-glucoside) were negatively correlated. The changes in correlations and contents were consistent.

The correlation between metabolites and unigene is represented as a network diagram (Figure 5). The above figure shows the correlation between metabolites and genes in flavonoid biosynthesis, anthocyanin biosynthesis, and isoflavonoid biosynthesis. The metabolites pinocembrin (dihydrochrysin), chlorogenic acid (3-O-caffeoylquinic acid), epiafzelechin, and afzelechin (3,5,7,4′-tetrahydroxyflavan), as well as the genes associated with them, played a significant role in flavonoid biosynthesis. The production of differential metabolites is likely related to these genes.

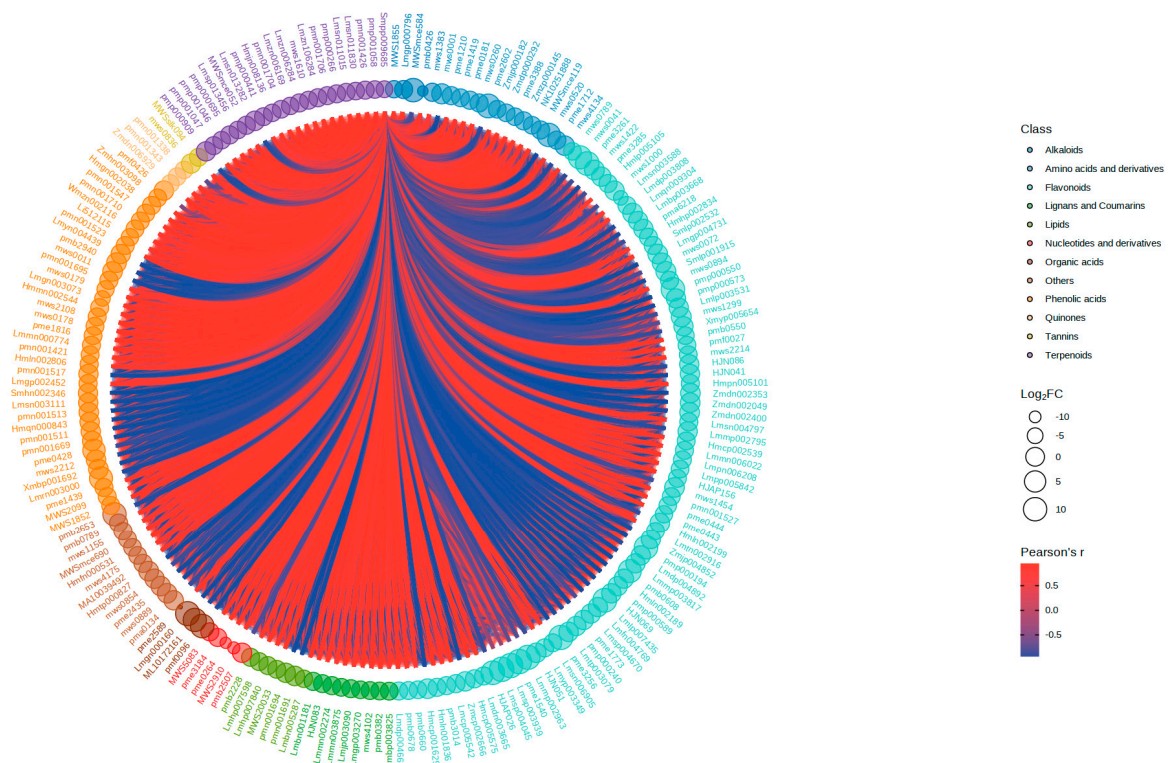

**Figure 4.** Differential metabolites and chord diagrams. The outermost number in the figure is the difference in metabolites. The size of the middle point represents the log$_2$FC value. The larger the point, the larger the corresponding log$_2$FC value. Blue lines represent a negative correlation and red lines represent a positive correlation.

**a Flavonoid Biosynthesis**  **b Anthocyanin Bisynthesis**  **c Isoflavonoid Biosynthesis**

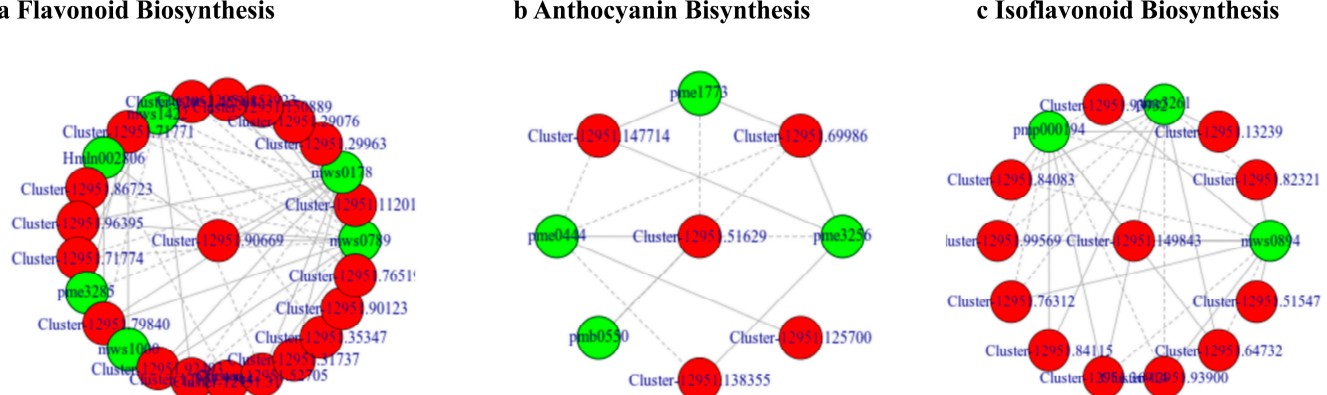

**Figure 5.** The correlation diagram of metabolites and genes is labelled green and the gene is marked red. The solid line represents a positive correlation and the dotted line represents a negative correlation. (**a**). Genetics and metabolites related to flavonoid biosynthesis. (**b**). Anthocyanin biosynthesis genes and metabolites. (**c**). Genes and metabolites related to isoflavonoid biosynthesis.

In the synthetic pathway of flavonoids (Figure 6), we found that pinocembrin (dihydrochrysin) in the flavanones, epiafzelechin and afzelechin (3,5,7,4′-tetrahydroxyflavan) in the flavanols, fustin in the flavanonols, malvidin-3-O-glucoside (Oenin), cyanidin-3-O-rutinoside (keracyanin) and delphinidin-3-O-rutinoside, 6″-O-malonylgenistin, and 6-hydroxydaidzein in MY content were higher than in MW content. Glycitin (GlyCitein 7-O-glucoside) and cyanidin-3-O-glucoside (kuromanin) had lower MWs than MYs.

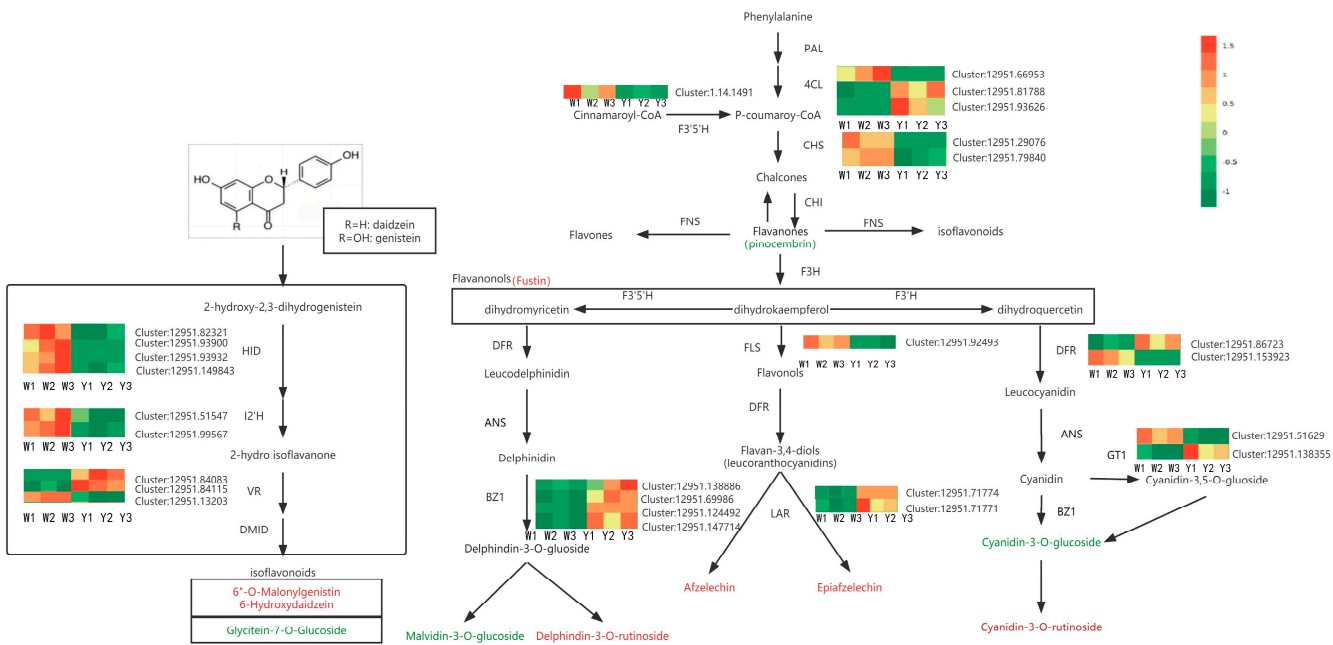

**Figure 6.** Significant diagram of flavonoid biological synthesis of *R. liliiflorum*. The shrinking words and expression patterns of enzymes are displayed next to each metabolic step and the synthetic direction is represented by a black arrow. The RNA-Seq expression mode of the gene is displayed in the heat map. The colour was obtained by the FPKM value converted from log2. Some data used to generate the gene heat maps are reported by [21].

In Figure 6, we found that CHS, FLS, DFR, LAR, BZ1, GT1, 4CL, HID, I2′H, VR, etc., played a role in connection. The correspondence between unigene and enzymes can be seen in Table S2. In the correlation between genes and metabolites (Figure 6), we found that the expression of these enzymes and pinocembrin (dihydrochrysin), epiafzelechin, afzelechin (3,5,7,4′-tetrahydroxyflavan), fustin, malvidin-3-O-glucoside (oenin), cyanidin-3-O-rutinoside (keracyanin), delphinidin-3-O-rutinoside, 6″-O-malonylgenistin, 6-hydroxydaidzein, cyanidin-3-O-glucoside (kuromanin), and glycitin (glycitein 7-O-glucoside) showed different positive and negative correlations. The different positive correlations presented by the enzyme unigene further verified that more than 10 metabolites were accumulated in MW and MY. The difference between the colour of the *R. liliiflorum* petal may be caused by the cumulative accumulation of more than 10 metabolites in MW and MY.

## 4. Discussion

Most rhododendrons are bright red, purple, etc. In the late stage of *R. liliiflorum*, the petals are composed of yellow and upper white colours at the base and contain ornamental value. By analysing the data of the metabolic group, we obtained the metabolites of *R. liliiflorum* petals and used the Kyoto genes and the genome of the encyclopedia (KEGG) database to comment on the two parts of *R. liliiflorum* yellow and white petals. Some of the differences in metabolites were significantly enriched in flavonoid biosynthesis (KO00941), anthocyanin biosynthesis (KO00942), and isoflavone biosynthesis (KO00943). In the above three biological biosynthesis pathways, only 11 different metabolites were enriched. The pinocembrin (dihydrochrysin) in flavanones, epiafzelechin and afzelechin (3,5,7,4′-tetrahydroxyflavan) in flavanols, fustin in flavanonols, cyanidin-3-O-glucoside (kuromanin), malvidin-3-O-glucoside (oenin), cyanidin-3-O-rutinoside (keracyanin) and delphinidin-3-O-rutinoside in anthocyanin biosynthesis, 6′-O-malonylgenistin, glycitein 7-O-glucoside, and 6-hydroxydaidzein were enriched in isoflavonoid biosynthesis. The above 11 substances are conditionally attributable to the colour differences caused in *R. liliiflorum*. In the previous transcription group article, we found that the genes related to CHS, F3′5′h, FLS, I2′H, HID, DFR, and LAR were key to the colour differences in *R. liliiflorum* petals [19].

In this article, we used widely targeted metabolomics to further reveal the reasons for the colour differences in *R. liliiflorum* petals.

### 4.1. Analysis of Metabolites and Related Enzymes Enriched in Flavonoid Biosynthesis

In synthetic biosynthesis, flavanones are very important intermediate substrates [33]. Flavanones can participate in the biosynthesis of chalcones, flavonoids, and isoflavones under the action of various enzymes [34]. Substances such as chalcones, isoflavones, and flavonoids have been reported in related documents to play a key role in the formation of yellow flowers [19,35]. Pinocembrin was detected in the skin of red and orange species in related studies of Sorghum bicolor. It was conditionally found that pinocembrin is the reason for the colour differences affecting sweet sorghum seeds [36]. Common forefronts of a flavanone compound are also a flavonol compound, anthocyanin, and procyanidins [37,38]. In this process, pinocembrin (dihydrochrysin), an important metabolite in dihydrogen flavonoids, will also convert to other metabolites [39]. The genes that show a correlation with pinocembrin are CHS, FLS, DFR, and LAR. CHS is the first rate-limiting enzyme in flavonoid biosynthesis, and FLS is a bridge between the synthesis path of flavonoids and the synthesis of catechins. The expression of its related genes will affect the biological synthesis of flavanones [40–44]. In this study, we found that CHS- and FLS-related genes were downregulated in the yellow part of *R. liliiflorum* (Figure 6). The pinocembrin (dihydrochrysin) content in the yellow petal part of *R. liliiflorum* also decreased. The most important thing was that the synthesis of the expression of CHS and FLS related to pinocembrin (dihydrochrysin) showed a positive relationship (Figure 5a).

Flavanonols are an important intermediate metabolic product and a key branch point in flavonoid biosynthesis [45]. The cumulative accumulation of flavanol compounds in different parts of the petals also causes colour differences [46]. Epiafzelechin and afzelechin (3,5,7,4′-Tetrahydroxyflavan) are flavanol compounds, and fustin is a flavananol. In Figure 6, we noticed that by participating in epiafzelechin, afzelechin mainly influenced the biological synthesis of flavonoids through CHS-, LAR-, and CHS-related enzymes and then regulated the synthesis of flavanonols and catechin synthesis downstream [40–42]. LAR-related genes form a flavanol compound by regulating leucocyanidin [47]. As the enzyme downstream of fustin, LAR participates in the regulation of fustin. The expression of LAR-related genes did not cause a decline in the content of fustin in the upstream flavanonols. This may have been due to competition between DFR and FLS for the common substrate flavanonols, as DFR and FLS suppress each other's transcription [48]. If LAR wants to convert flavanonols into a flavanol compound, first, under the action of DFR, the flavanonols are converted into leucocyanidin, and then flavanols are generated under the action of LAR [43,44]. The regulation of epiafzelechin, afzelechin, and fustin showed a positive relationship (Figure 4). This is consistent with the increase in the contents of epiafzelechin and afzelechin (3,5,7,4′-tetrahydroxyflavan) in flavanols in the yellow portion of the petals of *R. liliiflorum*.

### 4.2. Analysis of Metabolites and Related Enzymes Enriched in Isoflavonoid Biosynthesis

In isoflavonoid biosynthesis, the expression levels of isoflavones in yellow flowers were higher than those in white flowers [21,49]. In this study, the two isoflavone metabolites, 6′′-O-malonylgenistin and 6-hydroxydaidzein, were higher in the yellow part than in the white part of *R. liliiflorum*, and only the yellow part was detected. GlyCitin (glyCitein 7-O-glucoside) in the yellow part of *R. liliiflorum* was lower than that in the white part (Figure 6). Relevant studies have shown that glyCitin (glyCitein 7-O-glucoside) will form isoflavonoid compounds such as malonylglycitin [50]. This shows that in the process of regulating glycitin (glycitein 7-O-glucoside) synthesis, I2′H and VR may transform glycitin into other isoflavones, which may cause the glycitin (glycitein 7-O-glucoside) content of the yellow part of *R. liliiflorum* to be lower than that of the white part.

### 4.3. Analysis of Metabolites and Related Enzymes Enriched in Anthocyanin Biosynthesis

We believe that malvidin-3-O-glucoside (oenin), delphinidin-3-O-rutinoside, cyanidin-3-O-glucoside (kuromanin), and cyanidin-3-O-rutinoside (keracyanin) accumulation in different parts of *R. liliiflorum* may be a factor that leads to its colour differences. In related studies, anthocyanins have been found to be the most important types of colourful agents in flavonoid biosynthesis [51]. Their accumulation in different parts of the petals differentiate the colour of the petals [21,24]. Cyanidin-3-O-rutinoside (keracyanin) is found in yellow cherry and apricot. A higher accumulation of related anthocyanins increases the chances that the colour will change from yellow to red [52,53]. Figure 6 shows that GT1 and BZ1 are enzymes that regulate anthocyanins, and the expression of related genes may affect the accumulation of anthocyanin compounds [54,55].

### 5. Conclusions

This article used widely targeted metabolomics to study the differences in unigene expression between the yellow part of *R. liliiflorum* and the white part of the *R. liliiflorum* flower cuckoo. We compared the different metabolites in the KEGG database and found numerous differences in anthocyanin biosynthesis, flavonoid biosynthesis, and isoflavone biosynthesis. Different metabolites in the enrichment of flavonoid synthesis and isoflavone synthesis pathways are flavanonols (fustin), flavanols (epiafzelechin, afzelechin), flavanones (pinocembrin), and isoflavones (6″-O-malonylgenistin, glycitin, 6-hydroxydaidzein). The differences between each part of these seven substances in the yellow part of the *R. liliiflorum* petal and the white part may be a factor that causes the colour differences between the colours of *R. liliiflorum* petals. Additionally, we also found that malvidin-3-O-galactoside (primulin), delphinidin-3-O-rutinoside, cyanidin-3-O-glucoside (kuromanin), and cyanidin-3-O-rutinoside (keracyanin) may cause colour differences between the colours of *R. liliiflorum* petals. We also found that CHS, FLS, LAR, DFR, HID, and I2′H play a key role in *R. Liliiflorum* colour formation while using differential metabolite and unigene coanalysis. This study reveals the differential metabolites and genes involved in the biosynthesis pathways of flavonoids that contribute to the colour differences between the yellow and white parts of *R. liliiflorum*. These findings provide fundamental data for subsequent research on the regulation of flower colour in *R. liliiflorum* and have implications for the horticultural applications of *R. liliiflorum*.

**Supplementary Materials:** The online version contains supplementary material available at https://www.mdpi.com/article/10.3390/horticulturae9121351/s1: Figure S1: The main component analysis (PCA), orthogonal signal correction, and OPLS-DA model plots; Table S1: Differential flavonoid metabolites in the white and yellow fractions of *R. liliiflorum*; Table S2: The corresponding relationship between unigene and enzyme.

**Author Contributions:** Conceptualization, L.L. and T.S.; data collection, Q.Z. and X.M.; data anlysis, J.D. and X.W.; article writing, J.D. and X.W.; funding acquisition, X.Z. (Xu Zhang); supervision, X.Z. (Ximin Zhang), Z.Z. and Y.Y. All authors have read and agreed to the published version of the manuscript.

**Funding:** This research was funded by the Guizhou Provincial Science and Technology Projects [QIANKEHEJICHU-ZK [2021] Key 038, QIANKEHEZHICHENG [2022] Key 017, QIANKEHEPINGTAIRENCAI [2017] 5726-15]; Guizhou Provincial Basic Research Program (Natural Science) [QIANKEHEJICHU-ZK [2023] Key 268]; National Science Foundation of China NSFC [32260225]; Natural Science Foundation of China and the Karst Science Research Center of Guizhou Province, China (U1812401); Higher Education Science and Research Youth Project of Guizhou Education Department (Qianjiaoji [2022]130); and Key Laboratory of Environment Friendly Management on Alpine Rhododendron Diseases and Pests of Institutions of Higher Learning in Guizhou Province, Guizhou Normal University (Qianjiaoji [2022]044). The Research Foundation for Science & Technology Innovation Team of Guizhou Province (Grant No. QJJ[2023]063).

**Data Availability Statement:** The data that support the findings of this study are available from the corresponding author upon reasonable request. The data are not publicly available due to no public repository for raw data is provided.

**Conflicts of Interest:** The authors declare no conflict of interest.

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
