# Peer review of "UPLC–MS/MS and Gene Expression Research to Distinguish the Colour Differences of Rhododendron liliiflorum H. Lév"

_horticulturae, doi:10.3390/horticulturae9121351_

Round 1
Reviewer 1 Report
Comments and Suggestions for Authors
After reading the manuscript titled: UPLC‒MS/MS and gene expression research to distinguish the 2 color differences of Rhododendron liliiflorum H. Lév., by Jin Dai, Xinglin Wang, Xingpan Meng, Xu Zhang, Qihang Zhou, Zhengdong Zhang , Ximin Zhang, Yin Yi, Lunxian Liu, Yingliang Liu and Tie Shen, I consider the manuscript original and relevant to the academic community that researches the topic. I indicate some aspects that should be improved.
Abstract: Make the objective of the research clear in the summary.
Keywords: replace UPLC‒MS/MS.
Introduction: Line 66 and 69: ...Mizuta D et al.....;Sun et al. .......cite the publication number specified in the references.
Methodology:
There is information in the results that needs to be present in the material and method section.
Item 2.1. What is the geographic coordinate of the location where the species was collected?
Results:
This section needs to be revised as many sections are not results but rather material and method.
Lines 190 and 192 are part of the methodology and not the results.
Lines 217 and 226 replace Figure 2A and Figure 2B with Figures 2a and Figure 2b, the text description must be the same as that shown in the figure.
Line 267 Pearson correlations should be described in detail in the materials and methods section.
Line 293, figure 4 must be reordered, it does not follow what is described in the text.
lines 297 to 302 this section is not a result, but methodology.
References at the end of the manuscript must be reviewed, especially the name of the journals, in compliance with the standards of the journal in question.
Author Response
Dear Reviewers,
We thank you for the opportunity to revise the manuscript entitled “UPLC‒MS/MS and gene expression research to distinguish the colour differences of Rhododendron liliiflorum H. Lév” for publication in Horticulturae. We also thank the reviewers and you for the helpful comments. Following those suggestions, we have made a number of revisions to the manuscript as outlined below. We look forward to your response and hope the revisions will enable you to accept this version of the manuscript.In this revised version, changes to our manuscript were all highlighted within the document using red-colored text.
Comments:
“After reading the manuscript titled: UPLC‒MS/MS and gene expression research to distinguish the 2 color differences of Rhododendron liliiflorum H. Lév., by Jin Dai, Xinglin Wang, Xingpan Meng, Xu Zhang, Qihang Zhou, Zhengdong Zhang , Ximin Zhang, Yin Yi, Lunxian Liu, Yingliang Liu and Tie Shen, I consider the manuscript original and relevant to the academic community that researches the topic. I indicate some aspects that should be improved”
We also appreciate your clear and detailed feedback and hope that the explanation has fully addressed all of your concerns. In the remainder of this letter, we discuss each of your comments individually along with our corresponding responses.
To facilitate this discussion, we first retype your comments in italic font and then present our responses to the comments.
Comment 1: Abstract: Make the objective of the research clear in the summary.
Response 1: Thank you for pointing this out. We have already stated the objective of the research clearly in the abstract. You can find this change in the revised manuscript, specifically in lines 30-34.
Comment 2: Keywords: replace UPLC‒MS/MS.
Response 2: Thank you for pointing this out. We changed the keyword and used petal instead of UPLC-MS/MS. You can find this change in the revised manuscript, specifically in line 35.
Comment 3: Introduction: Line 66 and 69: ...Mizuta D et al.....; Sun et al. .......cite the publication number specified in the references.
Response 3: We agree with this comment. We have added citations after "Mizuta D et al.....; Sun et al." You can find this change in the revised manuscript, specifically in lines 70 and 73.
Comment 4: Methodology:
There is information in the results that needs to be present in the material and method section.
Item 2.1. What is the geographic coordinate of the location where the species was collected?
Response 4: Thank you for your comment. We have added the geographical coordinates of the Baili Rhododendron Nature Reserve in the northwest of Guizhou Province and the nursery at Guizhou Normal University. The geographical coordinates of the Baili Rhododendron Nature Reserve in the northwest of Guizhou Province are N 27°12′54″, E 105°55′5″, and the geographical coordinates of the nursery at Guizhou Normal University are N 26°35′18″, E 106°43′18″. This change is found in the revised manuscript, specifically in lines 109 and 111.
Comment 5: Results:
This section needs to be revised as many sections are not results but rather material and method.
Lines 190 and 192 are part of the methodology and not the results.
Response 5: Thank you for pointing this out. We have made modifications to the Results section by moving the content of Lines 190 and 192 to the Materials and Methods section. You can find this change in the revised manuscript, specifically in lines 178-179.
Comment 6: Lines 217 and 226 replace Figure 2A and Figure 2B with Figures 2a and Figure 2b, the text description must be the same as that shown in the figure.
Response 6: Thank you for pointing this out. We have replaced Figures 2A and 2B with Figures 2a and 2b, and have checked the rest of the text for any other revisions. You can find this change in the revised manuscript, specifically in lines 233 and 242.
Comment 7: Pearson correlations should be described in detail in the materials and methods section.
Response 7: Thank you for pointing this out. We have described the Pearson correlation in the Materials and Methods section. You can find this change in the revised manuscript, specifically in lines 172-174.
Comment 8: Line 293, figure 4 must be reordered, it does not follow what is described in the text.
Response 8: Thank you for pointing this outUpon review, we discovered that we had overlooked the labeling of Figure 4 in the text. We have now re-labeled Figure 4 to align with the corresponding section in the article. You can find this change in the revised manuscript, specifically in line 285.
Comment 9: lines 297 to 302 this section is not a result, but methodology.
Response 9: Thank you for pointing this out. We have modified the result part and changed the content of Lines 297 and 302 to the part of Materials and Methods. You can find this change in the revised manuscript, specifically in lines 179-184.
Comment 10: References at the end of the manuscript must be reviewed, especially the name of the journals, in compliance with the standards of the journal in question.
Response 10: Thank you for pointing this out. We have made changes to the journal names in the references by abbreviating the journal names. We have added the missing page numbers for the journal references. You can find this change in the revised manuscript.
We would like to take this opportunity to thank you for all your time involved and this great opportunity for us to improve the manuscript. We hope you will find this revised version satisfactory.
Sincerely,
Jin Dai, Xinglin Wang, Xingpan Meng, Xu Zhang, Qihang Zhou, Zhengdong Zhang, Ximin Zhang, Yin Yi, Lunxian Liu, and Tie Shen

Reviewer 2 Report
Comments and Suggestions for Authors
Dear Authors,
I have reviewed the manuscript, and I present my comments below.
The title and the manuscript are formally correct, but the textual publication of the Internet references is not correct, I ask the authors to check this formally.
The aim of the research is not properly described in the Abstract, I ask the authors to replace it.
The literature references used in the Introduction chapter are good, but these should be supplemented with literature references published in the last 5 years, because most of the literature sources used are much older than that. I am asking the authors to change the chapter.
The figures used are good, their quality is adequate, but I ask the authors that any in-text reference to the figure or any part of it should follow the figure, for the sake of better coherence, and that the montages should be compiled in such a way that when the authors refer to a part of that figure, within the montage, it should be in front. Thus, for example, in the case of Figure 1, point B is referred to first, even though it should be point a. I ask the authors to fix this.
The Discussion chapter is appropriate.
The Conclusion part is not good, I ask the authors to show the goal, in addition to the results of the article, and what they achieved with this result on a larger level, even globally and where the result fits, who and how they can use it, etc.
Author Response
Dear Reviewer,
We thank you for the opportunity to revise the manuscript entitled “UPLC‒MS/MS and gene expression research to distinguish the colour differences of Rhododendron liliiflorum H. Lév” for publication in Horticulturae. We also thank the reviewers and you for the helpful comments. Following those suggestions, we have made a number of revisions to the manuscript as outlined below. We look forward to your response and hope the revisions will enable you to accept this version of the manuscript.In this revised version, changes to our manuscript were all highlighted within the document using red-colored text.
We also appreciate your clear and detailed feedback and hope that the explanation has fully addressed all of your concerns. In the remainder of this letter, we discuss each of your comments individually along with our corresponding responses.
To facilitate this discussion, we first retype your comments in italic font and then present our responses to the comments.
Comment 1: The title and the manuscript are formally correct, but the textual publication of the Internet references is not correct, I ask the authors to check this formally.
Response 1: Thank you for pointing this out. We have made changes to the journal names in the references by abbreviating the journal names. We have added the missing page numbers for the journal references. You can see this change in the references section.
Comment 2: The aim of the research is not properly described in the Abstract, I ask the authors to replace it.
Response 2: Thank you for pointing this out. We have already stated the objective of the research clearly in the abstract. This change is found in the revised manuscript, specifically in lines 30-34.
Comment 3: The literature references used in the Introduction chapter are good, but these should be supplemented with literature references published in the last 5 years, because most of the literature sources used are much older than that. I am asking the authors to change the chapter.
Response 3: Agree. We have made changes to the articles in the introduction section by incorporating recent articles from the past 5 years. You can find this change in the revised manuscript, specifically in lines50, 62, 86, and 89.
Comment 4: The figures used are good, their quality is adequate, but I ask the authors that any in-text reference to the figure or any part of it should follow the figure, for the sake of better coherence, and that the montages should be compiled in such a way that when the authors refer to a part of that figure, within the montage, it should be in front. Thus, for example, in the case of Figure 1, point B is referred to first, even though it should be point a. I ask the authors to fix this.
Response 4: Thank you for pointing this out. Upon inspection, we discovered that there was an issue with the order of some figures due to our oversight. We have rectified this by changing the citation order of Figure 1, with Figure 1a being cited first, followed by Figure 1b. You can find this change in the revised manuscript, specifically in lines 107, 187, and 189.
Comment 5: The Conclusion part is not good, I ask the authors to show the goal, in addition to the results of the article, and what they achieved with this result on a larger level, even globally and where the result fits, who and how they can use it, etc.
Response 5: Thank you for pointing this out. We have made changes to the Results section by including a description of the objectives. You can find this change in the revised manuscript, specifically in lines 434-438.
We would like to take this opportunity to thank you for all your time involved and this great opportunity for us to improve the manuscript. We hope you will find this revised version satisfactory.
Sincerely,
Jin Dai, Xinglin Wang, Xingpan Meng, Xu Zhang, Qihang Zhou, Zhengdong Zhang, Ximin Zhang, Yin Yi, Lunxian Liu, and Tie Shen
